# S3TA: A Soft, Spatial, Sequential, Top-Down Attention Model

## Abstract

We present a soft, spatial, sequential, top-down attention model (S3TA). This model uses a soft attention mechanism to bottleneck its view of the input. A recurrent core is used to generate query vectors, which actively select information from the input by correlating the query with input- and space-dependent key maps at different spatial locations.

We demonstrate the power and interpretabilty of this model under two settings. First, we build an agent which uses this attention model in RL environments and show that we can achieve performance competitive with state-of-the-art models while producing attention maps that elucidate some of the strategies used to solve the task. Second, we use this model in supervised learning tasks and show that it also achieves competitive performance and provides interpretable attention maps that show some of the underlying logic in the model's decision making.

## 1 Introduction

Traditional RL agents and image classifiers rely on some combination of convolutional and fully connected components to gradually process input information and arrive at a set of policy or class logits. This sort of architecture is very effective, but does not lend itself to easy understanding of how decisions are made, what information is used and why mistakes are made. Previous efforts to visualize deep RL agents (Greydanus et al. (2017); Zahavy et al. (2016); Wang et al. (2015)) focus on generating saliency maps to understand the magnitude of policy changes as a function of a perturbation of the input. This can uncover some of the "attended" regions, but may be difficult to interpret. For example, it can't reveal certain types of behavior when the agent makes decisions based on components absent from a frame. Our mechanism provides a more direct interpretation by making the attention a core part of the network.

In this work we present a soft, spatial, sequential and top-down attention model (S3TA, pronounced SETA). This model enables us to build agents and classifiers that actively select important, task-relevant information from visual inputs by sequentially querying and receiving compressed query-dependent summaries to generate appropriate outputs. To do this, the model generates attention maps, which can uncover some of underlying decision process used to solve the task. By observing and analyzing the resulting attention maps we can make educated guesses at *how* the system solves a task and where and why it might be failing. In the RL domain, we observe that the attention focuses on the key components of each level: tracking the region ahead of the player, focusing on enemies and important moving objects. In supervised learning, we observed that the attention sequentially focuses on different portions of the input to build up confidence in a classification or resolve ambiguity between different class labels. We also find that our model maintains competitive performance on both learning paradigms while providing interpretability.

## 2 Model

Our model, outlined in Figure 1, **queries** a large input tensor through an attention mechanism and uses the returned compressed **answer** (a low dimensional summary of the input) to produce its output. We refer to this full query-answer system as an **attention head**. Our system can implement multiple attention heads by producing multiple queries and receiving multiple answers.

An observation $\mathbf{X} \in \mathbb{R}^{H \times W \times C}$ at time $t$ (here an RGB frame of height $H$ and width $W$) is passed through a "vision core". The vision core is a multi-layer convolutional network $\text{vis}_\theta$ followed by a recurrent layer with state $\boldsymbol{s}_{\text{vis}}(t)$ such as a ConvLSTM (Shi et al. (2015)), which produces an output tensor $\mathbf{O}_{\text{vis}} \in \mathbb{R}^{h \times w \times c}$:

$$\mathbf{O}_{\text{vis}}, \boldsymbol{s}_{\text{vis}}(t) = \text{vis}_\theta(\mathbf{X}(t), \boldsymbol{s}_{\text{vis}}(t-1)) \tag{1}$$

The vision core output is then split along the channel dimension into two tensors: the "Keys" tensor $\mathbf{K} \in \mathbb{R}^{h \times w \times C_k}$ and the "Values" tensor $\mathbf{V} \in \mathbb{R}^{h \times w \times C_v}$, with $c = C_V + C_K$. To the keys and values tensors we concatenate a spatial basis — a fixed tensor $\mathbf{S} \in \mathbb{R}^{h \times w \times C_S}$ which encodes spatial locations (see below for details).

A recurrent neural network (RNN) with parameters $\phi$ produces $N$ queries, one for each attention head. The RNN sends its state $\boldsymbol{s}_{\text{RNN}}$ from the previous time step $t - 1$ into a "Query Network". The query network $Q_\psi$ is a multi-layer perceptron (MLP) with parameters $\psi$ whose output is reshaped into $N$ query vectors $\boldsymbol{q}^n$ of size $C_k + C_S$ such that they match the channel dimension of $\mathbf{K}$:

$$\boldsymbol{q}^1 ... \boldsymbol{q}^N = Q_\psi(\boldsymbol{s}_{\text{RNN}}(t-1)) \tag{2}$$

Similar to Vaswani et al. (2017), we take the inner product between each query vector $\boldsymbol{q}^n$ and all spatial locations in the keys tensor $\mathbf{K}$ to form the $n$-th attention logits map $\tilde{\boldsymbol{A}}^n \in \mathbb{R}^{h \times w}$:

$$\tilde{A}^n_{i,j} = \sum_c q^n_c K_{i,j,c} \tag{3}$$

where $K \in \mathbb{R}^{h \times w \times C_k + C_S}$ is the concatenation along the channel dimension of $\mathbf{K}$ and $\mathbf{S}$. We then take the spatial softmax to form the final normalized attention map $\boldsymbol{A}^n$:

$$A^n_{i,j} = \frac{\exp(\tilde{A}^n_{i,j})}{\sum_{i,j} \exp(\tilde{A}^n_{i,j})} \tag{4}$$

Each attention map $\boldsymbol{A}^n$ is broadcast along the channel dimension of the values tensor $\mathbf{V}$, point-wise multiplied with it and then summed across space to produce the $n$-th answer vector $\boldsymbol{a}^n \in \mathbb{R}^{1 \times 1 \times C_v + C_s}$:

$$a^n_c = \sum_{i,j} A^n_{i,j} V_{i,j,c} \tag{5}$$

where $V \in \mathbb{R}^{h \times w \times C_v + C_S}$ is the concatenation along the channel dimension of $\mathbf{V}$ and $\mathbf{S}$ Finally, the $N$ answer vectors $\boldsymbol{a}^n$, and the $N$ query form the input to the RNN core to produce the next RNN state $\boldsymbol{s}_{\text{RNN}}(t)$ and output $\boldsymbol{o}(t)$ for this time step:

$$\boldsymbol{o}(t), \boldsymbol{s}_{\text{RNN}}(t) = \text{RNN}_\phi(a^1, ..., a^n, q^1, ..., q^n, \boldsymbol{s}_{\text{RNN}}(t-1)) \tag{6}$$

The exact details for each of the networks, outputs and states are given in Section 4 and the Appendix.

It is important to emphasize several points about the proposed model. First, the model is fully differentiable due to the use of soft-attention and can be trained using back-propagation. Second, the query vectors are a function of the RNN core state alone and not the observation — this allows for a "top-down" mechanism where the RNN can actively query the input for task-relevant information rather than having to filter out large amounts of information. Third, the spatial sum (equation 5) is a severe spatial bottleneck, which forces the system to make the attention maps in such a way that information is not "blurred" out during summation.

The summation of the values tensor of shape $h \times w \times C_v$ to an answer of shape $1 \times 1 \times C_v$ is invariant to permutation of spatial position, which emphasizes the need for the spatial basis. Due to the spatial structure being lost during the spatial summation, the only way the RNN core can know and reason about spatial positions is by using the channels coming from the spatial basis [1]. We postulate that the query and answer structure can have different "modes" — the system can ask "where" ("what") something is by sending out a query with zeros in the spatial channels of the query and non-zeros in the channels corresponding to the keys (which are input dependent). It can then read the answer from the spatial channels, localizing the object of interest. Conversely it can ask "what is in this

---

[1] The vision core might have some ability to produce information regarding absolute spatial positioning, but due to its convolutional structure is limited.

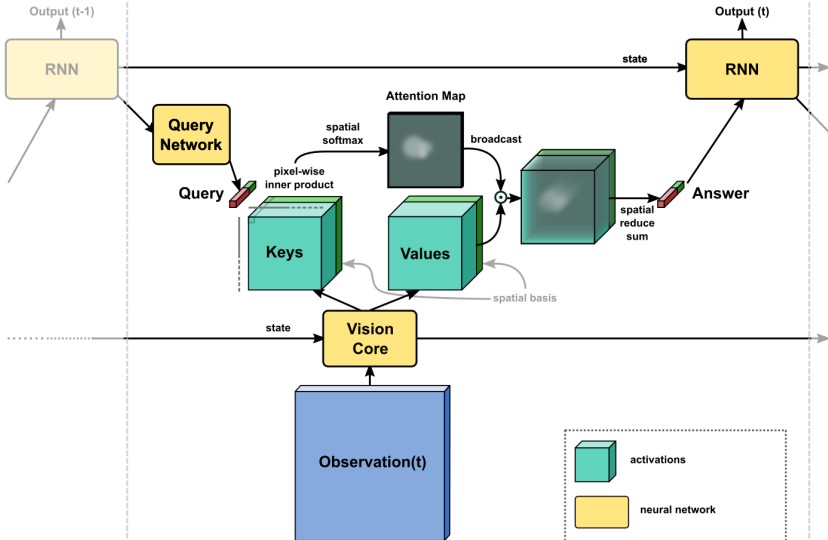

Figure 1: An outline of our proposed model. Observations pass through a (recurrent) vision core network, producing a "keys" and a "values" tensor, to both of which we concatenate a spatial basis tensor (see text for details). A recurrent network at the top sends its state from the previous time-step into a query network which produces a set of query vectors (only one is shown here for brevity). We calculate the inner product between each query vector and each location in the keys tensor, then take the spatial softmax to produce an attention map for the query. The attention map is broadcast along the channel dimension, point-wise multiplied with the values tensor and the result is then summed across space to produce an answer vector. This answer is sent to the top RNN as input to produce the output and next state of the RNN.

particular location" by zeroing out the content channels of the query and putting information on the spatial channels, reading the content channels of the answer and ignoring the spatial channels. This is not a dichotomy as the two can be mixed (e.g. "find enemies in the top left corner"), but it does point to an interesting "what" and "where" separation, which we discuss in Section 4.1.5.

## 2.1 THE SPATIAL BASIS

The spatial basis $\mathbf{S} \in \mathbb{R}^{h \times w \times C_S}$ such that the channels at each location $i, j$ encode information about the spatial position. Adding this information into the values of the tensor allows some spatial information to be maintained after the spatial summation (equation 5) removes the structural information. Following Vaswani et al. (2017) and Parmar et al. (2018) we use a Fourier basis type of representation. Each channel $(u, v)$ of $\mathbf{S}$ is an outer product of two Fourier basis vectors. We use both odd and even basis functions with several frequencies. For example, with two even functions one channel of $\mathbf{S}$ with spatial frequencies $u$ and $v$ would be:

$$S_{i,j,(u,v)} = \cos(\pi u i/h) \cos(\pi v j/w) \tag{7}$$

where $u, v$ are the spatial frequencies in this channel, $i, j$ are spatial locations in the tensor and $h, w$ are correspondingly the height and width of the tensor. We produce all the outer products such that the number of channels in $\mathbf{S}$ is $(U + V)^2$ where $U$ and $V$ are the number of spatial frequencies we use for the even and odd components (4 for both throughout this work, so 64 channels in total).

The spatial basis can also be learned as another parameter of the model — while we tested this in some cases we did not observe that this makes a big difference in performance and for brevity this is not done in this work.

## 3 RELATED WORK

There is a vast literature in recurrent attention models. They have been applied with some success to question-answering datasets (Hermann et al., 2015), text translation (Vaswani et al., 2017; Bahdanau

et al., 2014), video classification and captioning (Shan & Atanasov, 2017; Li et al., 2017), image classification and captioning (Mnih et al., 2014; Chung & Cho, 2018; Fu et al., 2017; Ablavatski et al., 2017; Xiao et al., 2015; Zheng et al., 2017; Wang et al., 2017; Xu et al., 2015; Ba et al., 2014), text classification (Yang et al., 2016; Shen & Lee, 2016), generative models (Parmar et al., 2018; Zhang et al., 2018; Kosiorek et al., 2018), object tracking (Kosiorek et al., 2017), and reinforcement learning (Choi et al., 2017). These attention mechanisms can be grouped by whether they use hard attention (e.g. Mnih et al. (2014); Ba et al. (2014); Malinowski et al. (2018)) or soft attention (e.g. Bahdanau et al. (2014)) and whether they explicitly parameterize an attention window (e.g. Jaderberg et al. (2015); Shan & Atanasov (2017)) or use a weighting mechanism (e.g. Vaswani et al. (2017); Hermann et al. (2015)).

Our work introduces a novel architecture which builds on existing methods. We use a soft key, query, and value type of attention similar to Vaswani et al. (2017) and Parmar et al. (2018), but instead of doing "self"-attention where the queries come from the input (together with the keys and values) we have a different, top-down source for them. This enables the system to be both state/context dependent and input dependent. Furthermore the output of the attention model is highly compressed and has no spatial structure (other than the one preserved using the spatial basis), unlike in "self" attention where each pixel attends to every other pixel and the structure is preserved. Finally, we apply the attention sequentially in time similar to Xu et al. (2015) but with a largely different attention mechanism.

Of existing models, the MAC model (Hudson & Manning, 2018) is the closest to ours. There are several differences between our model and MAC. First, MAC was built to solve CLEVR (Johnson et al., 2017); major parts of it are geared for that dataset. Specifically the "control" unit is built to expect a guiding question for the reasoning process — this may not always exist, such as in the case of RL or classic supervised learning where the systems needs to come up with its own queries to produce the required output. Another difference is the use of a pre-trained ResNet-101 (Wang et al., 2017) as the visual backend; we train the visual core to co-adapt with the top-down mechanism such that it learns to produce useful keys and values for different queries. Finally, MAC does not use a spatial basis. It can still reason about space to some extent through the fully connected layers, but there is not a clear separation between space and content as in our model.

## 4 ANALYSIS AND RESULTS

### 4.1 REINFORCEMENT LEARNING

We use the Arcade Learning Environment (Bellemare et al. (2013b)) to train and test our agent on 57 different Atari games.

For this experiment, the model uses a 3 layer convolutional neural network followed by a convolutional LSTM as the vision core. The RNN is an LSTM that generates a policy $\pi$ and a baseline function $V^\pi$; it takes as input the query and answer vectors, the previous reward and a one-hot encoding of the previous action. The query network is a three layer MLP, which takes as input the hidden state $h$ of the LSTM from the previous time step and produces 4 attention queries. See Appendix A.1.1 for a full specification of the network sizes.

We use the Importance Weighted Actor-Learner Architecture (Espeholt et al. (2018)) training architecture to train our agents. We use an actor-critic setup and a VTRACE loss with an RMSProp optimizer (see learning parameters in Appendix A.1.1 for more details).

We compare against two models without bottlenecks to benchmark performance, both using the deeper residual network described in Espeholt et al. (2018). In the Feedforward Baseline, the output of the ResNet is used to directly produce $\pi$ and $V^\pi$, while in the LSTM Baseline an LSTM with 256 hidden units is inserted on top of the ResNet. The LSTM also gets as input the previous action and previous reward. We find that our agent is competitive with these state-of-the-art baselines, see Table 1 for benchmark results and Appendix A.1.3 for learning curves and performance on individual levels. Our model provides an attention map which shows the parts of space which are attended to by each attention head. This gives us hints as to what information from the input is used when producing the output. Though these do not necessarily tell the whole story of decision making,

---

[1] All referenced videos can be found at https://sites.google.com/view/s3ta.

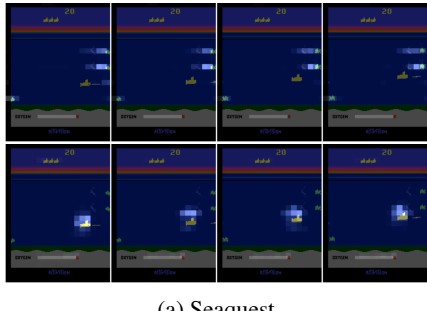

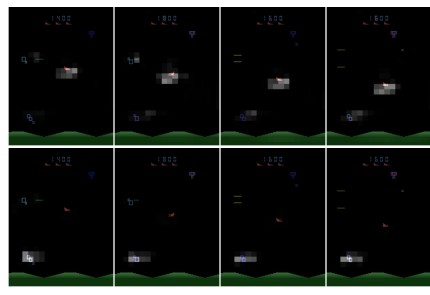

| (a) Seaquest | (b) Star Gunner |

Figure 2: Basic attention patterns. Bright areas are regions of high attention. Here we show 2 of the 4 heads used (one head in each row, time goes from left to right). The model learns to attend key sprites such as the player and different enemies. Best viewed on a computer monitor. See text for more details.

they do expose some of the strategies used by the model to solve the different tasks. Here we present some of these strategies and their relationship to the task at hand. Additionally, we analyze the use of the spatial basis vs. keys in the queries as a first step towards understanding the "what" and "where" in the system. We note that all the strategies we discuss here have been observed in more than one game or task; they are reproducible across multiple runs and we postulate they are effective strategies for the solution of the task at hand.

In order to visualize the attention maps we show the original input frame and super-impose the attention map $A^n$ for each head on it using alpha blending. This means that the bright areas in all images are the ones which are attended to, darker areas are not. We find the range of values to be such that areas which are not attended have weights very close to zero, meaning that little information is "blended" from these areas during the summation in equation 5. A more detailed analysis of the distribution of weights can be seen in Appendix A.2.1.

### 4.1.1 THE ROLE OF TOP-DOWN INFLUENCE

To test the importance of the top-down queries, we train two additional agents with modified attention mechanisms that do not receive queries from the top-level RNN but are otherwise identical to our agent. The first agent uses the same attention mechanism except that the queries are a learnable bias tensor which does not depend on the LSTM state. The second agent does away with the query mechanism entirely and forms the weights for the attention by computing the L2 norm of each key (similar to a soft version of Malinowski et al. (2018)). Both of these modifications turn the top-down attention into a bottom-up attention, where the vision network has total control over the attention weights.

We train these agents on 7 ATARI games for $2e9$ steps and compare the performance to the agent with top-down attention. We see significant drops in performance on 6 of the 7 games. On the remaining game, Seaquest, we see substantially improved performance; the positions of the enemies follow a very specific pattern, so there is little need for sequential decision making in that environment. On these games we see a median human normalized score of $541.1\%$ for the attention agent, $274.7\%$ for the fixed-query agent, and $274.5\%$ for the L2-Norm Key Agent. Mean scores are $975.5\%$, $615.2\%$ and $561.0\%$ respectively. See Appendix A.1.4 for more details.

Table 1: Human normalized scores for experts on ATARI.

| Model | Median | Mean |
|---|---|---|
| Feedforward Baseline | 284.5% | 1479.5% |
| LSTM Baseline | 45.0% | 1222.0% |
| Attention | **407.1%** | **1649.0%** |

### 4.1.2 BASIC ATTENTION PATTERNS

The most dominant pattern we observe is that the model learns to attend to task-relevant things in the scene. In most ATARI games that usually means that the player is one of the focii of attention, as well as enemies, power-ups and the score itself (which is an important factor in the calculating the value function). Figure 2 (best viewed on screen) shows several examples of these attention maps. We also recommend watching the videos posted online for additional visualizations.

### 4.1.3 FORWARD PLANNING/SCANNING

In games where there is an element of forward planning and a direct mapping between image space and world space (such as 2D top-down view games) we observe that the model learns to scan through possible paths emanating from the player character and going through possible future trajectories. Figure 3 shows a examples of this in Ms Pacman and Alien — in the both games the model scans through possible paths, making sure there are no enemies or ghosts ahead. We observe that when it does see a ghost, another path is produced or executed in order to avoid it. Again we refer the reader to the videos for a better impression of the dynamics.

### 4.1.4 "TRIP WIRES"

In many games we observe that the agent learns to place "trip-wires" at strategic points in space such that if something crosses them a specific action is taken. For example, in Space Invaders two such trip wires are following the player ship on both sides such that if a bullet crosses one of them the agent immediately evades them by moving towards the opposite direction. Another example is Breakout where we can see it working in two stages. First the attention is spread out around the general area of the ball, then focuses into a localized line. Once the ball crosses that line the agent moves towards the ball. Figure 4 shows examples of this behavior.

### 4.1.5 "WHAT" VS. "WHERE"

As discussed in Section 2, each query has two components: one interacts with the keys tensor - which is a function of the input frame and vision core state - and the other interacts with the fixed spatial basis, which encodes locations in space. Since the output of these two parts is added together via an inner product prior to the softmax, we can analyze, for each query and attention map, which part of the query is more responsible for the the attention at each point; we can contrast the "what" from the "where". For example, during a game a query may be trying to find ghosts or enemies in the scene, in which case the "what" component should dominate as these can reside in many different places. Alternatively, a query could ask about a specific location in the screen (e.g., if it plays a special role in a game), in which case we would expect the "where" part to dominate.

In order to visualize this we color code the relative dominance of each part of the query. When a specific location is more influenced by the contents part, we will color the attention red, and when it is more influenced by the spatial part, we color it blue. Intermediate values will be white. More details can be found in Appendix A.2.

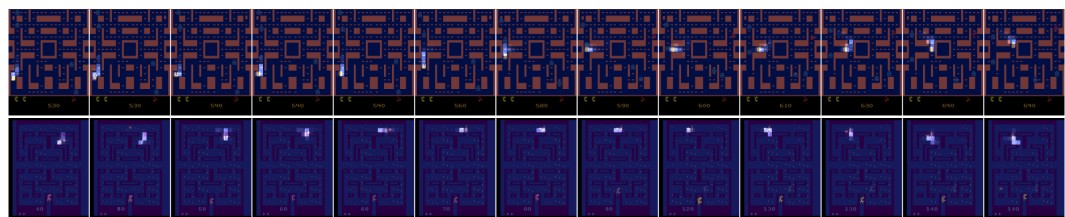

Figure 3: Forward planning/scanning. We observe that in games where there is a clear mapping between image space and world space and some planning is required, the model learns to scan through possible future trajectories for the player and chooses ones that are safe/rewarding. The images show two such examples from Ms Pacman and Alien. Note how the paths follow the map structure. See text for more details and videos. Bright areas are regions of high attention.

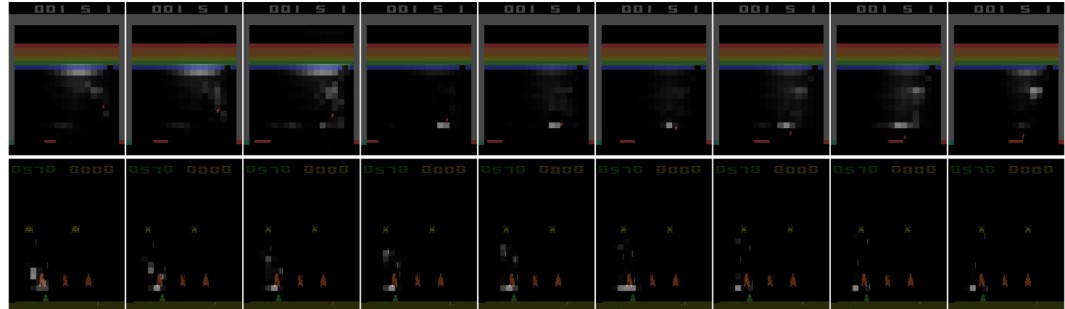

Figure 4: Trip Wires. We observe in games where there are moving balls or projectiles that the agent sets up tripwires to create an alert when the object crosses a specific point or line. The agent learns how much time it needs to react to the moving object and sets up a spot of attention sufficiently far from the player. In Breakout (top row), one can see a two level tripwire: initially the attention is spread out, but once the ball passes some critical point it sharpens to focus on a point along the trajectory, which is the point where the agent needs to move toward the ball. In Space Invaders (bottom row) we see the tripwire acting as a shield; when a projectile crosses this point the agent needs to move away from the bullet. Bright areas are regions of high attention.

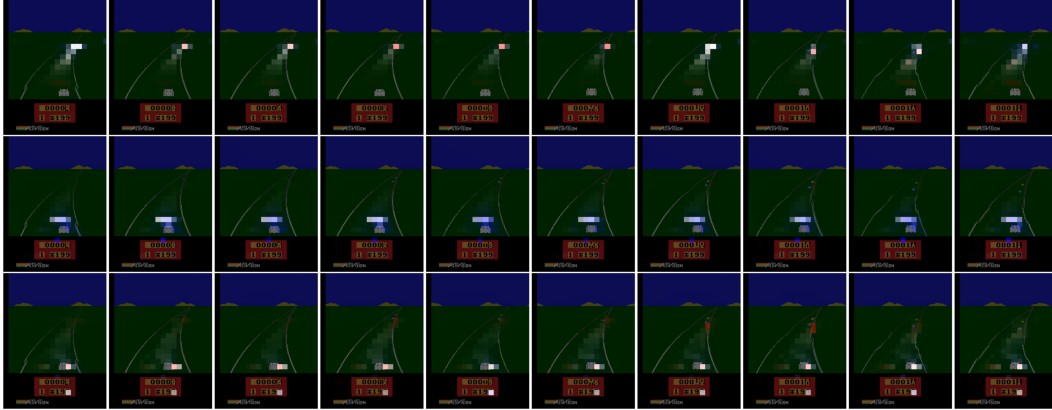

Figure 5: What/Where. This figures shows a sequence of 10 frames from Enduro (arranged left-to-right) along with the what-where visualization of each of the 3 of the 4 attention heads. (stacked vertically). The top row is the input frame at that timestep. Below we visualize the relative contribution of "what" vs. "where" in different attention heads: Red areas indicate the query has more weight in the "what" section, while blue indicates the mass is in the "where" part. White areas indicate that the query is evenly balanced between what and where. We notice that the first head here scans the horizon for upcoming cars and then starts tracking them (swithing from mixed to "what"). The second head is mostly a "where" query following the car for upcoming vehicles (a "trip-wire"). The last head here mostly tracks the player car and the score (mostly "what").

Figure 5 shows several such maps $C$ visualized in Enduro for different query heads. As can be seen, the system uses the two modes to make its decisions, some of the heads are content specific looking for opponent cars. Some are mixed, scanning the horizon for incoming cars and when found, tracking them, and some are location based queries, scanning the area right in front of the player for anything the crosses its path (a "trip-wire" which moves with the player). Examples of this mechanism in action can be seen in the videos online.

### 4.1.6 COMPARISON WITH OTHER ATTENTION ANALYSIS METHODS

In order to demonstrate that the attention masks are an accurate representation of where the agent is looking in the image, we perform the saliency analysis presented in Greydanus et al. (2017) on both the attention agent and the baseline feedforward agent. This analysis works by introducing a small, local Gaussian blur at a single point in the image and measuring the magnitude of the change in the policy. By measuring this at every pixel in the image, one can form a response map that shows how much the agent relies on the information at every spatial point to form its policy.

To produce these maps we run a trained agent for $> 200$ unperturbed frames on a level and then repeatedly input the final frame with perturbations at different locations. We form two saliency maps $S_\pi(i,j) = 0.5||\pi(\mathbf{X}'_{i,j}) - \pi(\mathbf{X})||^2$ and $S_{V^\pi}(i,j) = 0.5||V^\pi(\mathbf{X}'_{i,j}) - V^\pi(\mathbf{X})||^2$ where $\mathbf{X}'_{i,j}$ is the input frame blurred at point $(i,j)$, $\pi$ are the softmaxed policy logits and $V^\pi$ is the value function. An example of these saliency maps is shown in Figure 6. We see that the saliency map (in green) corresponds well with the attention map produced by the model and we see that the agent is sensitive to points in its planned trajectory, as we discussed in Section 4.1.3. Furthermore we see the heads specialize in their influence on the model — one clearly affects the policy more where the other affects the value function.

Comparing the attention agent to the baseline agent, we see that the attention agent is sensitive to more focused areas along the possible future trajectory. The baseline agent is more focused on the area immediately in front of the player (for the policy saliency) and on the score, while the attention agent focuses more specifically on the path the agent will follow (for the policy) and on possible future longer term paths (for the value).

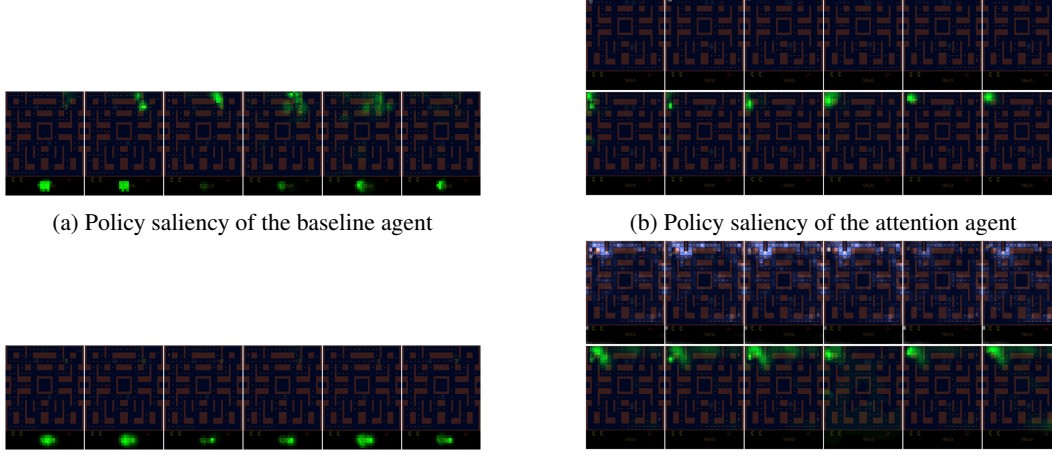

(a) Policy saliency of the baseline agent

(b) Policy saliency of the attention agent

(c) Value saliency of the baseline agent

(d) Value saliency of the attention agent

Figure 6: Saliency analysis. We run saliency analysis (see text for details) for the policy and value functions for both ours and the baseline feedforward agent. We visualize saliency in green, and in the case of our model the attention weights in white. We find that in the attention agent, one can see that the policy saliency (b) corresponds to the head that is most focused on the immediate actions of Pacman, while the the value saliency (d) corresponds to the head that is looking further ahead (two scales of planning/scanning behaviour). Comparing the saliency of the baseline and attention agents, the attention agent exhibits sharper saliency, which looks along specific paths and follows the contours of the map. The saliency of the baseline agent (a, c) shows the network is concerned with shorter timescales and uses the score as the most important input to the value function (in some frames the value function does look at the map, but the majority of the time it is focused on the scene). See text for details and videos.

## 4.2 SUPERVISED LEARNING

We test the S3TA mechanism on several image and video classification problems to explore its applicability to other tasks. For image classification, we present the image to the network multiple times, allowing the model to ask new queries of the same image as a function of the previous class logits.

### 4.2.1 IMAGENET

For ImageNet classification, the model needs substantially more capacity than it does for reinforcement learning. For the vision core, we use a 50-layer ResNet (He et al. (2016)) with no recurrent layer (since there is no motion to process). On top of the ResNet we use a 3-layer MLP to produce

the class logits at each timestep. The output logits are accumulated across time, adding the output of the MLP to the current logits. The Query network is a 4-layer MLP that takes as input the previous (accumulated) logits. The cross-entropy loss is applied to the accumulated class logits at the final timestep.

We ran several baselines, including a standard ResNet 50-layer model. We also create a recurrent version of this model by using a shared, 1-layer MLP to transform each time step's logits into a 224x224 tensor that is then added to the image at the next time step.

For our model, we find that accuracy initially improves as a function of the number of tiling steps. Our best result is for sequence length of eight timesteps and achieves 74.5% top-1, 91.5% top-5 accuracy. This is an improvement of 0.5% top-1 and 0.4% top-5 over a single ResNet-50 trained with our setup. Our findings are summarized in Table 2.

Table 2: Performance on ImageNet Test Dataset

| Model | Top-1 | Top-5 |
|---|---|---|
| Resnet-50 (He et al. (2016)) | 75.6% | 92.9% |
| Resnet-50 (our setup) | 74.0% | 91.1% |
| Resnet-50, Sequence Length 4 | 70.2% | 88.6% |
| Attention + Resnet-50, Sequence Length 1 | 73.1% | 90.1% |
| Attention + Resnet-50, Sequence Length 4 | 73.4% | 91.0% |
| Attention + Resnet-50, Sequence Length 8 | 74.5% | 91.5% |

For ImageNet, S3TA initially attends to low-level edges (mostly around the contour of the object). It will then reduce the class choices under consideration by focusing on high-level features. In the case of dogs, the attention maps first identify that a type of dog is present; correspondingly, the class probabilities will be distributed across possible dog breed choices. The model will then focus on ears, faces, snouts and other distinctive features to tell the specific breed apart, producing peaked logits. An example of this is shown in Figure 7.

The model can alter its classification decisions midway through a sequence, even when it appears to be very confident. When dealing with occlusions, the model will use other image properties to gather relevant class context. An example of this is show in Figure 8. This shows the model is able to perform meaningful sequential computation that significantly alter its classification choices.

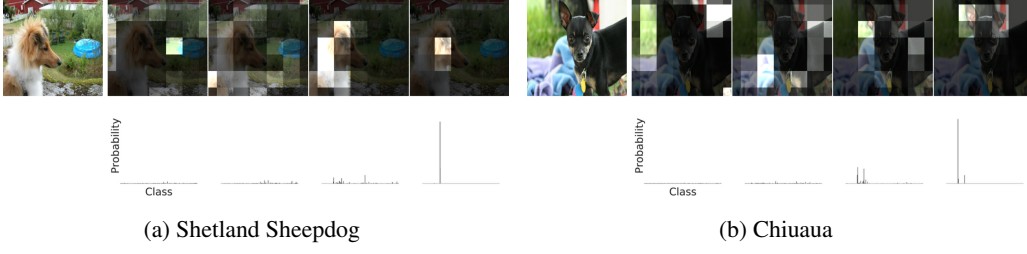

(a) Shetland Sheepdog                    (b) Chiuaua

Figure 7: ImageNet classification on two dog images from ImageNet. The input image is tiled four times. From left to right, the top row shows the input image then the four attention steps. The bottom row shows the corresponding logit outputs at each timestep. By the third frame, the model is sure both images are dogs, as indicated by similar class probability distributions. The attention snaps to specific patches in the last frame to discern the specific dog breed.

### 4.2.2 KINETICS

Kinetics is an action recognition video dataset where the goal is to classify videos portraying different actions correctly. We ran our model on the September 2018 version of the Kinetics 600 dataset (Carreira et al., 2018). For this model our vision core is a 34 layer ResNet followed by a convolutional LSTM; the rest of the model is identical to the ImageNet model. The videos in the dataset consist of 256 frames, from which we select 32 equally spaced frames to be processed sequentially

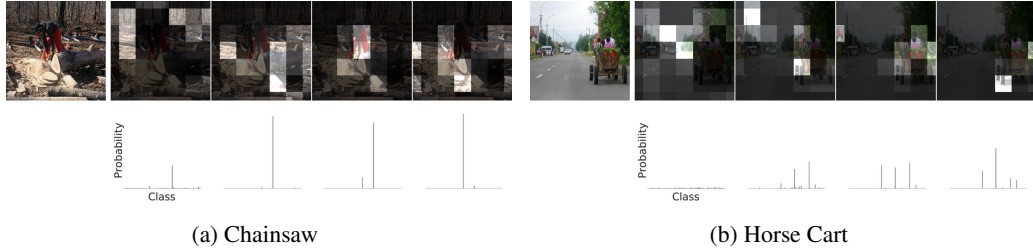

(a) Chainsaw            (b) Horse Cart

Figure 8: Confusion on ImageNet. In the first image, the tree-filled background initially makes S3TA suspect the class is "lumbermill". However, lumbermills are buildings full of mechanical items. The attention in the final frame focuses solely on the chainsaws, which become its final class choice. In the second, the horse is occluded in this image, and so S3TA has to use other clues to distinguish between "shopping cart", "barrow", and "horse cart". In the last frame, the attention maps focus on the horse whip on the right and the wheel type.

by the model. As before, the class logits are accumulated across the sequence and the last one is used as the output. We achieve 58% top-1, 82% top-5 accuracy on this dataset. The state-of-the-art (Carreira et al., 2018) achieves 71.7% top-1 accuracy, 90.4% top-5 accuracy.

In the case of the Kinetics dataset, the attention model often refrains from making a class prediction until a key item appears in the video sequence. The attention maps then focus on this object while it remains in view. For instance, the attention focuses on the musical instrument a person is playing, and the policy logits the narrow down to a few probable choices. If an action sequence is a sport, then the focus is typically on the game ball. Figure 9 shows an example of this behavior.

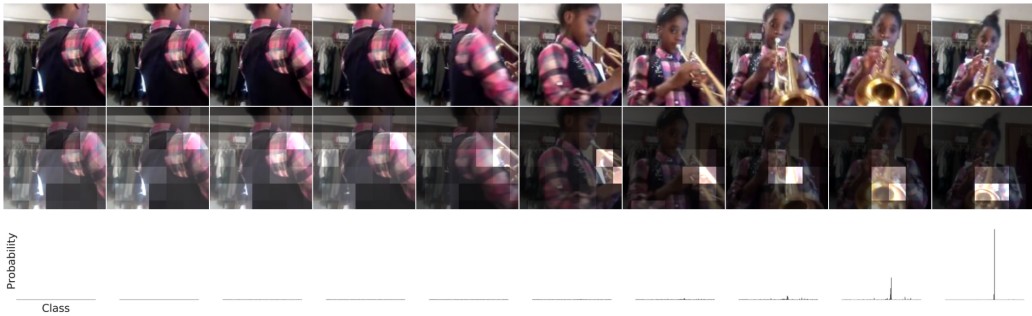

Figure 9: Focus on Key Items. The attention maps are disperse until a trumpet appears in view, at which point the class logits become very peaked. Bright areas are regions of high attention.

## 5   CONCLUSION

We have introduced S3TA, a model for sequential spatial top-down attention. This model learns to query its input for task-relevant information and receive spatially bottlenecked answers. The model performs well on a variety of RL and supervised learning tasks while providing some interpretabilty of its reasoning process.

The attention mechanism produces attention maps which can be used to visualize which parts of the input are attended to. We have seen that the agent is able to make use of a combination of "what" and "where" queries to select both regions and objects within the input depending on the task. In RL agents, we have seen that the agents are able to learn to focus on key features of the inputs, look ahead along short trajectories, and place tripwires to trigger certain behaviors. In supervised models, the model sequentially focuses on important parts of the model to build up confidence in its classification, and will hold off narrowing down its decision until key pieces of information become available. In both the RL and supervised learning paradigms, the model yields interpretable results without sacrificing performance.

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

# A   APPENDIX

## A.1   ATARI AGENT

### A.1.1   AGENT DESCRIPTION

Our agent takes in ATARI frames in RGB format ($210 \times 160 \times 3$) and processes them through a two layer ConvNet and a ConvLSTM, which produces an output of size $27 \times 20 \times 128$. We split this output along the channel dimension to produce keys of size $27 \times 20 \times 8$ and values of size $27 \times 20 \times 120$. To each of these we append the same spatial basis of size $27 \times 20 \times 64$. The query is produced by feeding the state of the LSTM after the previous time step to a three layer MLP. The final layer produces a vector with length 288, which is reshaped into a matrix of size $4 \times 72$ to represent the different attention heads. The queries, keys and values are processed by the mechanism described in Section 2 and produces answers. The queries, answers, previous action, and previous reward are fed into an answer processor, which is a 2 layer MLP. The output of the answer processor is the input to the policy core, which is an LSTM. The output of the policy core is processed through a one layer MLP and the output of that is processed by two different one layer MLPs to produce the policy logits and values estimate. All the sizes are summarizes in Table 3.

| Module | Type | Sizes |
|---|---|---|
| vision core | CNN | kernel size: $8 \times 8$, stride: 4, channels: 32
kernel size: $4 \times 4$, stride: 2, feature layers: 64 |
| vision RNN | ConvLSTM | kernel size: $3 \times 3$, channels: 128 |
| answer processor | MLP | hidden units: 512
hidden units: 256 |
| policy core | LSTM | hidden units: 256 |
| query network | MLP | hidden units: 256
hidden units: 128
hidden units: $72 \times 4$ |
| policy & value output | MLP | hidden units: 128 |

Table 3: The network sizes used in the attention agent

We an RMSProp optimizer with $\epsilon = 0.01$, momentum of 0, and decay of 0.99. The learning rate is $2e - 4$. We use a VTRACE loss with a discount of 0.99 and an entropy cost of 0.01 (described in Espeholt et al. (2018)); we unroll for 50 timesteps and batch 32 trajectories on the learner. We clip rewards to be in the range $[-1, 1]$, and clip gradients to be in the range $[-1280, 1280]$. Since the framerate of ATARI is high, we send the selected action to the environment 4 times without passing those frames to the agent in order to speedup learning. Parameters were chosen by performing a hyperparameter sweep over 6 levels (battle zone, boxing, enduro, ms pacman, seaquest, star gunner) and choosing the hyperparameter setting that performed the best on the most levels.

### A.1.2   MULTI-LEVEL AGENTS

We also train an agent on all ATARI levels simultaneously. These agents have distinct actors acting on different levels all feeding trajectories to the same learner. Following Espeholt et al. (2018), we train the agent using population based training (Jaderberg et al. (2017)) with a population size of 16, where we evolve the learning rate, entropy cost, RMSProp $\epsilon$, and gradient clipping threshold. We initialize the values to those used for the single level experts, and let the agent train for $2e7$ frames before begining evolution. We use the mean capped human normalized score described in Espeholt et al. (2018) to evaluate the relative fitness of each parameter set.

### A.1.3   AGENT PERFORMANCE

Figure 10 shows the training curves for the experts on 55 ATARI levels (the curves for Freeway and Venture are omitted since they are both constantly 0 for all agents). Table 1 shows the final human-

normalized score achieved on each game by each agent in both the expert and multi-agent regime. As expected, the multi-level agent achieves lower scores on almost all levels than the experts.

### A.1.4 TOP-DOWN VERSUS BOTTOM-UP

Figure 11 shows the training curves for the Fixed Query Agent and the L2 Norm Keys agent. These agents are all trained on single levels for $2e9$ frames. We see that, in 6 of the 7 tested games, the agents without top-down attention perform significantly worse than the agent with top-down attention. Table 5 shows the final scores achieved by each agent on all 7 levels.

### A.2 WHAT-WHERE ANALYSIS

To form the what-where maps shown in Section 4.1.5, we compute the relative contribution $C_{i,j}$ for a query $q$ from the content and spatial parts at each location is defined to be:

$$\text{what}_{i,j} = \sum_{h=1}^{C_k} q_h K_{i,j,h} \tag{8}$$

$$\text{where}_{i,j} = \sum_{h=1}^{C_s} q_{h+C_k} S_{i,j,h} \tag{9}$$

$$D_{i,j} = \begin{cases} -log(10) & \text{what}_{i,j} - \text{where}_{i,j} < -log(10) \\ \text{what}_{i,j} - \text{where}_{i,j} & |\text{what}_{i,j} - \text{where}_{i,j}| \leq log(10) \\ log(10) & \text{what}_{i,j} - \text{where}_{i,j} > log(10) \end{cases} \tag{10}$$

$$C_{i,j} = D_{i,j} A_{i,j} \tag{11}$$

where we interpolate between red, white and blue according to the values of $C$. The intuition is that, at blue (red) points the contribution from the spatial (content) portion to the total weights would be more than 10 times greater than the other portion. We truncate at $\pm 10$ because there are often very large differences in the logits, but after the softmax huge differences become irrelevant. We weight by the overall attention weight to focus the map only on channels that actually contribute to the overall weight map.

### A.2.1 ATTENTION WEIGHTS DISTRIBUTION

Since the sum that forms the attention answers (Equation 5) runs over all space, the peakiness of the attention weights will have a direct impact on how local the information received by the agent is. Figure 12 shows the distribution of attention weights for a single agent position in Ms Pacman and Space Invaders on all four heads. On both games we observe that some of the heads are highly peaked, while others are more diffuse. This indicates that the agent is able to ask very local queries as well as more general queries. It is worth noting that, since the sum preserves the channel structure, it is possible to avoid washing out information even with a general query by distributing information across different channels.

### A.3 SUPERVISED LEARNING

### A.3.1 IMAGENET

Table 6 summarizes the architecture we use to train on ImageNet.

We used a momentum-based optimizer with momentum = 0.9. The learning rate started at $1e-1$ for tile lengths 1 and 4; it set to $1e-2$ for tile length 8. Our batch size was size 1024, and we annealed the learning rate by $0.1$ at iterations 1.0e5, 1.5e5, and 1.75e5. We used a learning decay rate of $1e-4$. For training, we applied a data augmentation pipeline involving aspect ratio  color distortion as well as flipping the image horizontally. For the sequence length 8 result, we initially train with sequence length 4 for the first $1e5$ iterations and then switch to sequence length 8 for the remainder of the training. This greatly improves the training efficiency of the model.

### A.3.2 KINETICS 600

Table 7 contains the layer types and sizes we use to train on Kinetics.

As with our ImageNet experiments, we used a momentum-based optimizer with momentum = 0.9. The learning rate was set to $1e - 2$ and annealed at iterations 1.5e5, 2.0e5. Our batch size was of size 240.

Our training  testing pipelines are very close to those described in (Carreira et al., 2018). However, we don't pad videos to be of size 251 frames. Rather than employing their sampling procedure, we extract 32 frames in fixed intervals from each video.

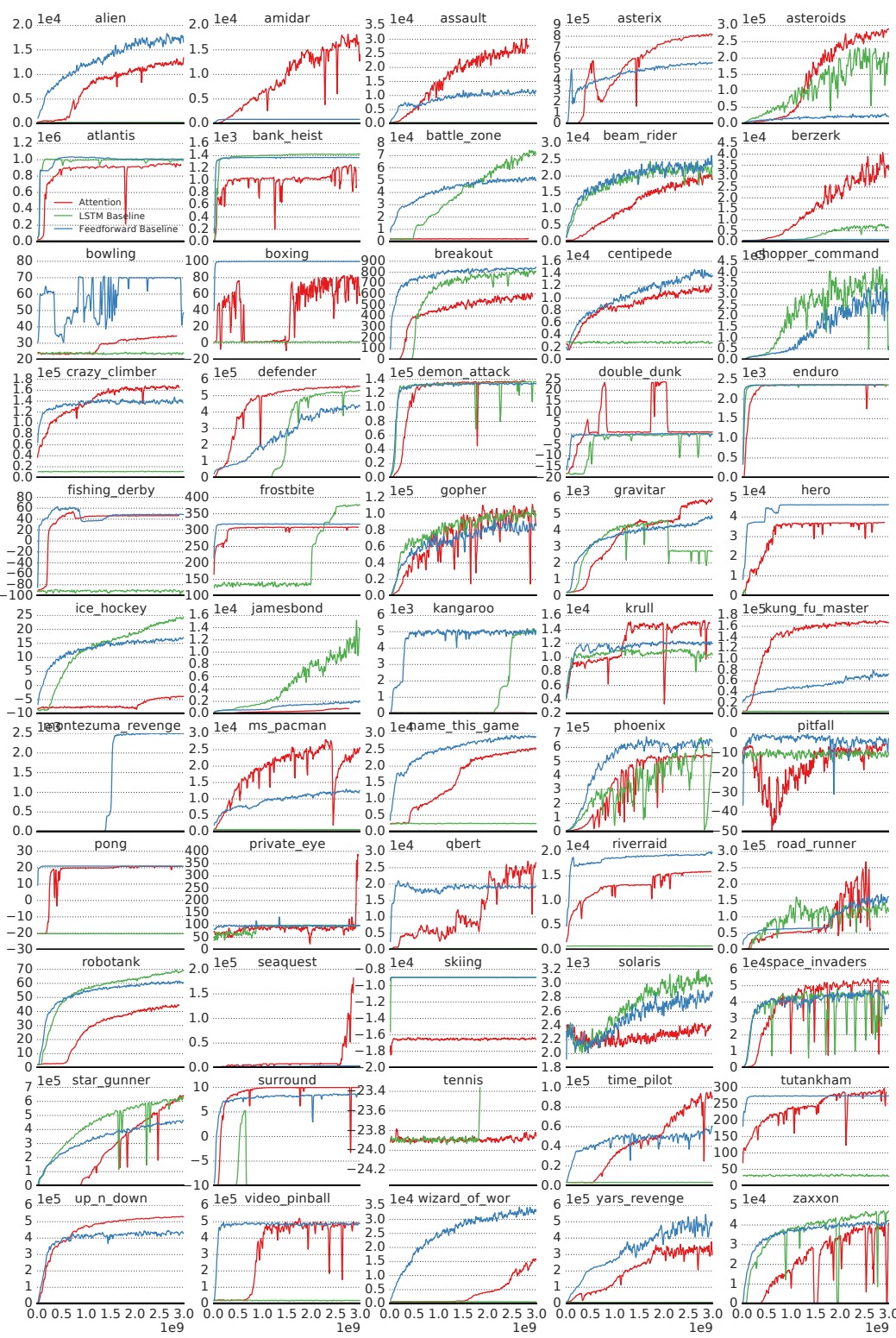

Figure 10: Performance of individual experts on selected ATARI games. `Freeway` and `Venture` are omitted; no tested agent achieved a non-zero return on either game

| Level | Experts | | | Multi-level | |
|---|---|---|---|---|---|
| | **Feedforward** | **LSTM** | **Attention** | **Feedforward** | **Attention** |
| alien | **271.8%** | 0.3% | 206.9% | 26.8% | 27.1% |
| amidar | 50.9% | 2.7% | **1138.9%** | 12.5% | 15.9% |
| assault | 2505.8% | 26.2% | **6571.9%** | 80.3% | 69.5% |
| asterix | 6827.5% | 0.7% | **9922.0%** | 14.2% | 29.5% |
| asteroids | 75.3% | 545.8% | **626.3%** | 1.6% | 2.7% |
| atlantis | **6320.7%** | 6161.6% | 5820.0% | 194.8% | 136.4% |
| bank_heist | 184.0% | **191.8%** | 168.5% | 4.2% | 1.7% |
| battle_zone | 151.9% | **216.2%** | 2.1% | 5.6% | 2.6% |
| beam_rider | **172.3%** | 152.1% | 132.7% | 1.8% | 1.4% |
| berzerk | 39.8% | 353.6% | **1844.3%** | 10.4% | 12.1% |
| bowling | **35.1%** | 1.7% | 9.0% | 3.8% | 3.1% |
| boxing | **832.5%** | 25.2% | 743.6% | 677.1% | 32.5% |
| breakout | **2963.5%** | 2917.4% | 2284.2% | 15.0% | 29.2% |
| centipede | **136.5%** | 12.7% | 108.3% | 43.1% | 35.4% |
| chopper_command | 5885.2% | **8622.1%** | 12.3% | 20.8% | 5.3% |
| crazy_climber | 560.7% | 5.6% | **643.9%** | 374.3% | 398.0% |
| defender | 2835.5% | 3361.2% | **3523.9%** | 98.9% | 76.9% |
| demon_attack | 7406.6% | 7526.0% | **7563.3%** | 47.4% | 112.5% |
| double_dunk | 865.2% | 850.8% | **1934.0%** | 108.4% | 171.6% |
| enduro | **275.0%** | 274.5% | 275.0% | 127.7% | 51.7% |
| fishing_derby | **293.9%** | 8.6% | 280.8% | 132.3% | 10.0% |
| freeway | 0.1% | 0.1% | 0.1% | **75.9%** | 12.9% |
| frostbite | 6.0% | 7.3% | 5.7% | **35.1%** | 4.7% |
| gopher | 4588.1% | 5124.6% | **5280.3%** | 36.4% | 141.6% |
| gravitar | 151.8% | 144.6% | **184.6%** | 3.8% | 3.1% |
| hero | **151.9%** | 6.7% | 121.7% | 43.2% | 22.2% |
| ice_hockey | 241.0% | **302.2%** | 64.1% | 37.7% | 35.6% |
| jamesbond | 845.9% | **5819.2%** | 319.7% | 31.7% | 13.0% |
| kangaroo | **178.9%** | 174.1% | 0.6% | 21.7% | 8.5% |
| krull | 1031.8% | 921.0% | **1309.6%** | 547.4% | 883.3% |
| kung_fu_master | 363.7% | 20.6% | **763.9%** | 73.3% | 118.1% |
| montezuma_revenge | **52.6%** | 0.1% | 0.1% | 0.0% | 0.1% |
| ms_pacman | 195.9% | 6.4% | **442.8%** | 31.6% | 26.4% |
| name_this_game | **482.3%** | 7.5% | 413.1% | 74.0% | 53.9% |
| phoenix | **10705.9%** | 10423.9% | 8560.2% | 47.5% | 63.3% |
| pitfall | **3.4%** | 3.4% | 3.4% | 3.4% | 3.4% |
| pong | **118.1%** | 2.0% | 118.1% | 55.3% | 2.1% |
| private_eye | 0.2% | 0.2% | 1.0% | 0.5% | **2.0%** |
| qbert | 160.6% | 1.2% | **207.7%** | 4.7% | 5.7% |
| riverraid | **118.6%** | -3.3% | 93.4% | 33.8% | 30.9% |
| road_runner | 2441.2% | 2336.6% | **3570.9%** | 409.7% | 284.8% |
| robotank | 625.3% | **700.3%** | 450.3% | 25.6% | 32.1% |
| seaquest | 8.5% | 0.6% | **546.5%** | 1.9% | 1.4% |
| skiing | 63.6% | 63.6% | 8.7% | **63.6%** | 63.4% |
| solaris | 15.7% | **19.1%** | 13.0% | 12.5% | 12.8% |
| space_invaders | 3230.4% | 3412.5% | **3668.0%** | 16.8% | 30.4% |
| star_gunner | 4972.8% | 6707.6% | **6838.6%** | 8.4% | 10.4% |
| surround | 114.2% | 93.0% | **121.9%** | 4.8% | 0.7% |
| tennis | **307.4%** | 153.5% | 0.7% | 49.8% | 45.4% |
| time_pilot | 3511.7% | 16.7% | **5708.4%** | 6.8% | 17.0% |
| tutankham | 169.3% | 19.3% | **187.3%** | 104.1% | 76.9% |
| up_n_down | 4035.0% | 12.3% | **4771.5%** | 347.8% | 59.1% |
| venture | 0.0% | 0.0% | 0.0% | **8.9%** | 3.1% |
| video_pinball | 2853.2% | 139.0% | **3001.8%** | 153.3% | 188.7% |
| wizard_of_wor | **842.5%** | 7.6% | 401.1% | 16.6% | 8.5% |
| yars_revenge | **1100.1%** | 12.7% | 867.0% | 47.8% | 32.2% |
| zaxxon | 472.2% | **521.1%** | 488.6% | 25.5% | 2.8% |

Table 4: The human-normalized score of agents on all ATARI levels.

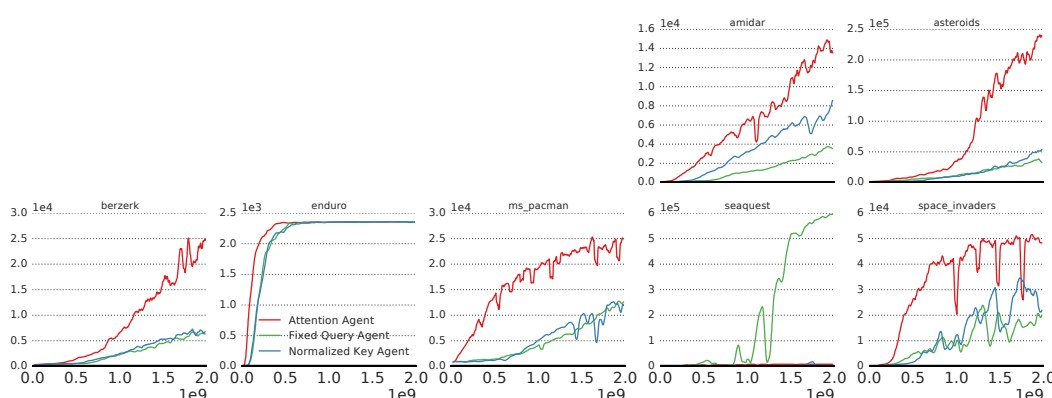

Figure 11: Performance of individual experts on selected ATARI games. `Freeway` and `Venture` are omitted; no tested agent achieved a non-zero return on either game

| level name | Fixed Query Agent | L2 Norm Keys Agent | Top-Down Attention Agent |
|---|---|---|---|
| `amidar` | 225.7% | 547.5% | **903.6%** |
| `asteroids` | 88.0% | 126.4% | **541.1%** |
| `berzerk` | 285.3% | 334.1% | **1153.9%** |
| `enduro` | 274.8% | 274.5% | **274.7%** |
| `ms_pacman` | 198.4% | 199.6% | **414.3%** |
| `seaquest` | **1435.9%** | 49.4% | 28.2% |
| `space_invaders` | 1798.1% | 2395.2% | **3512.8%** |

Table 5: The scores of the attention agent compared to the two bottom-up experiments described in the text.

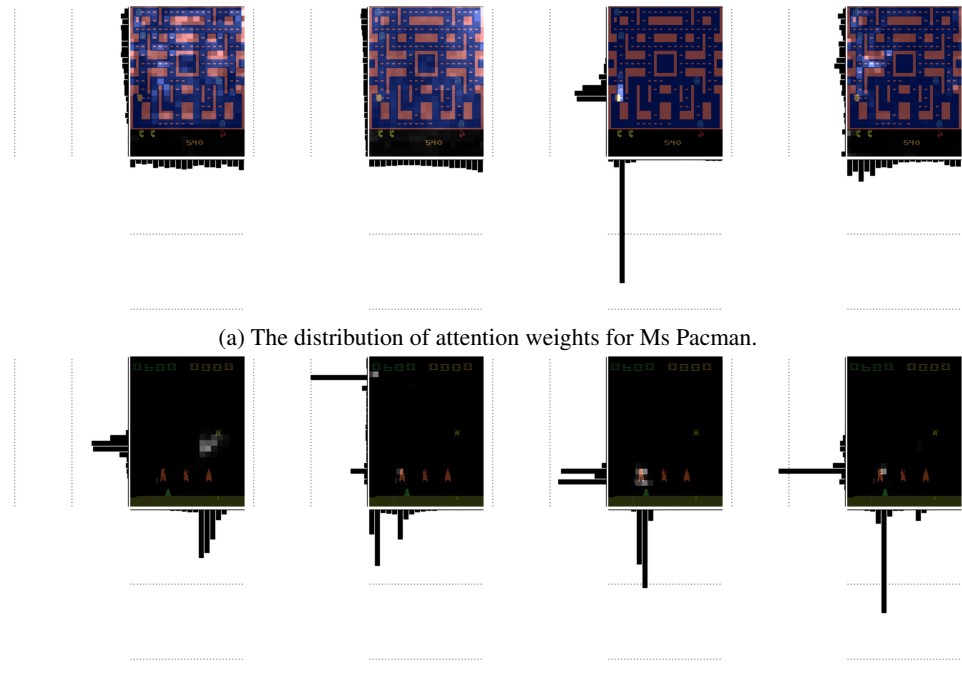

(a) The distribution of attention weights for Ms Pacman.

(b) The distribution of attention weights for Space Invaders

Figure 12: The distribution of attention weights on each head for a Ms Pacman and a Space Invaders frame. The two bar plots show the sum of the weights along the x and y axis (the range of each plot is [0, 1].

| Module | Type | Sizes |
|---|---|---|
| vision core | CNN | ResNet-50 (v2) |
| policy core | MLP | hidden units: 2048
hidden units: 2048
hidden units: 2048 |
| query network | MLP | hidden units: 1024
hidden units: 512
hidden units: 256
hidden units: 128 |

Table 6: The network sizes used in the ImageNet model.

| Module | Type | Sizes |
|---|---|---|
| vision core | CNN | ResNet-34 (v2) |
| vision | ConvLSTM | kernel size: $3 \times 3$, channels: 256 |
| policy core | MLP | hidden units: 1024
hidden units: 1024
hidden units: 1024 |
| query network | MLP | hidden units: 512
hidden units: 256
hidden units: 128 |

Table 7: The network sizes used in the Kinetics600 model.

