# OpenReview forum: "S3TA: A Soft, Spatial, Sequential, Top-Down Attention Model"
_ICLR.cc/2019/Conference_

### Official Review · AnonReviewer2 · 2018-11-02
**Interesting results. Some more experimentation needed**

**Rating:** 5
**Confidence:** 4

**Review:**

This work presents a recurrent attention model as part of an RNN-based RL framework. The attention over the visual input is conditioned on the the model's state representation at time t. Notably, this work incorporated multiple attention heads, each with differing behavior.

Pros:
-Paper was easy to understand
-Detailed analysis of model behavior. The breakdown analysis between "what" and "where" was particularly interesting.
-Attention results appear interpretable as claimed

Cons:
-Compared to the recurrent mechanism in MAC, both methods generate intermediate query vectors conditioned on previous model state information. I would not consider the fact that MAC expects a guiding question to initialize its reasoning steps constitute a major difference in the overall method.
-There should be an experiment demonstrating the effect of # of attention heads against model performance. How necessary is it to have multiple heads? At what point do we see diminishing returns?
-I would also recommend including a citation for :
Sukhbaatar, Sainbayar, Jason Weston, and Rob Fergus. "End-to-end memory networks." NIPS 2015.


General questions:
-Was there an effect of attention grid coarseness on performance?
-For the atari experiments, is a model action sampled after each RNN iteration? If so, would there be any benefit to trying multiple RNN iterations between action sampling?

---

> ### Author Response · Authors · 2018-11-16
> **Response**
>
> Thank you for your kind review and interest in the model, here is our response to concerns raised:
>
> * Relation to MAC
> As we mentioned, MAC is indeed close to this architecture. In terms of differences, first we would say that the existence of the question as guidance to the controller in MAC is a fundamental difference mainly because it serves as a very strong top-down bias for the vision (read) module - it’s not just in the initialization of the reasoning, but throughout the whole process. In the RL case you could think of the reward as having a somewhat related role, but we feel this is a big difference. Second, our model is trained end-to-end so the vision can adapt and produce keys/values which are task relevant - MAC uses a pre-trained ImageNet module (though we don’t argue MAC can’t be learned end-to-end, it’s just that it wasn’t, and in the RL tasks we use, it’s not clear what pre-trained network would be useful). Lastly, MAC does not use a spatial basis of any sort, so it can’t reason about absolute positioning as well as limited relative positioning (trip wires, for example, wouldn’t be possible without this).
>
> * Number of attention heads
> We observe that having only a single head produces substantially lower scores across a number of levels on ATARI and other environments.  We also see noticeable dropoff in performance in several levels beyond 8 heads.  Beyond that the answer is strongly level-dependent.  For example, an agent with 2 heads takes 200M steps more to reach maximum performance than one with 4 heads on Enduro, but an agent with 4 heads performs exactly the same as one with 8 heads.  In Berzerk, agents with 2, 4, and 8 heads all achieve similar scores. We chose 4 heads on the basis that 4 heads was never worse than 2 and occasionally better, while 8 heads was always very similar in performance to 4 heads.
>
> * Missing citation
> We will add this to the next version.
>
> * Coarseness of attention
> We checked on ImageNet, and the performance difference was negligible. The granularity isn’t particularly important for this dataset because the objects to classify are quite large and there’s no need to have a particularly fine level of attention.
>
> * Sampling of actions after every RNN iteration
> We sample a model action after every iteration.  We did not try to run the RNN multiple times before sampling the action, but our general intuition is that the frame rate in ATARI is high enough that this is generally not needed.  Indeed, we effectively sample an action after every four frames, because our model uses action repeat to speed up training (this is mentioned briefly in appendix A.1.1).

---

### Official Review · AnonReviewer1 · 2018-11-02
**An interesting visual attention approach.**

**Rating:** 5
**Confidence:** 4

**Review:**

Summary.
The paper proposes a variant model of existing recurrent attention models. The paper explores the use of query-based attention, spatial basis, and multiple attention modules running in parallel. The effectiveness of the proposed method is demonstrated with various tasks including RL (on Atari games), ImageNet image classification, and action recognition, and shows reasonable performance.

Strengths.
- An interesting problem in the current CV/RL community.
- Well-surveyed related work.
- Supplemental materials and figures were helpful in understanding the idea.

vs. Existing recurrent attention models.
In Section 2, the proposed model is explained with emphasizing the differences from existing models, but there needs a careful clarification.

In this paper, attention weights are computed conditioned on a query vector (which solely depends on the RNN’s state) and the Keys (which are generated by a visual encoder, called vision core). In the landmark work by Xu et al. (ICML ‘15, as already referenced), attention weights are computed very similarly - they used the hidden state of RNN followed by an additional layer (similar to the “query”) and visual features from CNN followed by an additional layer (similar to the “keys”). The only difference seems the use of element-wise multiplication vs. addition, but both are common units in building an attention module. Can authors clarify the main difference from the existing recurrent attention models?

Training details.
In the supervised learning tasks, are these CNN bases (ResNet-50 and ResNet-34) trained from scratch or pre-trained with another dataset?

Missing comparison with existing attention-based models.
The main contribution claimed is the attention module, but the paper does not provide any quantitative/qualitative comparison from another attention-based model. This makes hard to determine its effectiveness over others. Notable works may include:

[1] Sharma et al., “Action recognition using visual attention,” ICLR workshop 2016.
[2] Sorokin et al., “Deep Attention Recurrent Q-network,” NIPS workshop 2015.
[3] Choi et al., “Multi-Focus Attention Network for Efficient Deep Reinforcement Learning,” AAAI workshop 2017.

Minor concerns.
The related work section would be helpful if it proceeds the current Section 2.
Typos

---

> ### Author Response · Authors · 2018-11-16
> **Response**
>
> Thank you for you interest in the model and constructive comments. Below is our response.
>
> * Regarding Xu et al:
>
> It is true that our attention mechanism shares a number of features with Show, Attend, Tell.  The main differences between our model and SAT are:
> 1. We produce separate keys and values from the image.  In SAT, the attention weights are produced from the a vectors and the hidden state, and then the result is the weighted sum of the a vectors.  In our setup, the weights are produced from the k vectors and the hidden state, and then the result is the weighted sum of the v vectors.  Our method could, in principle, allow the system to specialize the information used to compute the weights compared to that used to produce the answers.  Furthermore, we are able to introduce an information asymmetry between the keys and values: the keys have only 8 channels, while the values have 120 channels, which is not possible in the SAT system
>
> 2. The addition of the spatial basis allows the system to ask positional questions.  The analysis in section 4.1.5 suggest that (on some games, at least) the agents are actively using the ability to ask for position-dependent information.  The inclusion of the spatial basis in the answer allows the system to learn the location within the image of the feature it just queried for and then ask a subsequent question about features around that location.  This can be seen in Figure 5, where the attention puts a location-based probe in the area where a new car will appear, and the content-based channels light up when the car does indeed appear.  This would be very difficult to do without the spatial basis, since it would have to encode a unique feature for a location that has no distinct features other than its position in space.
>
> 3. We compute multiple independent queries at each timestep.  This allows the system to track multiple threads at the same time.  This probably is not a vital feature for tasks such as image captioning, but for agents acting in an environment it is quite important.
>
> 4. While not a direct difference in the mechanism, it’s also worth emphasizing that the application of this sort of mechanism to the RL domain is novel and a substantial difference from the work presented in Xu et al.
>
> As you point out	, however, the attention mechanisms are quite similar and we will modify the text accordingly to reflect this.
>
>
> * Regarding pre-training of the ResNets:
> All CNNs are trained from scratch alongside the rest of the model.
>
> * Comparisons to existing attention-based models:
> [1] Sharma et al., “Action recognition using visual attention,” ICLR workshop 2016.
>
> While a direct comparison here would be difficulty due to use of a different dataset, we can observe that the resulting attention maps in this work are much blurrier and diffuse than the ones we get with our model.
>
> [2] Sorokin et al., “Deep Attention Recurrent Q-network,” NIPS workshop 2015.
>
> Compared to Sorokin, our attention mechanism provides much sharper attention maps.  Our final scores are substantially better, but it is hard to differentiate between the effect of our IMPALA-style training and the differences in the attention mechanism.
>
> [3] Choi et al., “Multi-Focus Attention Network for Efficient Deep Reinforcement Learning,” AAAI workshop 2017.
>
> Looking at the details in this paper, it appears that their attention mechanism is very similar to the “fixed query” baseline from section 4.1.1. They segment and process the image to arrive at a set of keys and values in an analogous way to the vision stack in our model.  The selectors from their paper are analogous to our learnable bias tensors.  There are differences: we use a ConvLSTM in the visual processing, a spatial basis, and an LSTM in the policy processing, but none of these affect the core attention mechanism.

---

### Official Review · AnonReviewer3 · 2018-11-05
**Interesting model, however, the performance on the supervised task is not good.**

**Rating:** 5
**Confidence:** 4

**Review:**


[Summary]

This paper proposed a soft, spatial, sequential, top-down attention model which enable the agent and classifier actively select important, task relative information to generate the appropriate output. Given the observations, the proposed method uses a convLSTM to produce the key and value tensor. Different from multi-head attention, the query vector is generated in a top-down fashion. The authors proposed to augment the spatial feature with Fourier bases which is similar to previous work. The authors verify the proposed method on both reinforcement learning and supervised learning. On reinforcement learning, the proposed method outperformed the feedforward baseline and LSTM baseline. On reinforcement learning task, the proposed method achieves compelling result with more interpretable attention map that shows the model's decision.

[Strength]
1: The proposed model is a straightforward extension of the multi-head attention to visual input. Compare to multi-head attention, it generates the query vector in a top-down manner instead of pure bottom up, and the authors verify the proposed choice is better than LSTM baseline empirically.

2: The authors verify the proposed method by extensive experiments on reinforcement learning tasks and also try supervised learning tasks. The attention visualization and human normalized scores for experts on ATARI show the effectiveness of the proposed method.

[Weakness]
1: The soft spatial top-down attention is very common in vision and language domain, such as VQA. As the authors mentioned, the proposed method is very similar with MAC for CLEVER. The sequential attention is also explored in previous VQA work. Thus the novelty of the proposed method is limited.

2: Multi-head attention for NLP tasks are usually composed with multiple layers. Will more layer of attention help the performance? The paper is less of this ablation study.

3: The proposed method is worse compared with other baselines on supervised learning tasks, on both imagenet classification and kinetics. I wonder whether the recurrent process is required for those tasks? On table 2, we can observe that with sequence length 8, the performance is much worse,  this may be caused by overfitting.

4: If the recurrent attention is more interpretable, given other visualization methods, such as gradcam, I wonder what is advantage?

5: I would expect that the performance on Kinetics dataset is better since sequential attention is required on video dataset. However, the performance is much worse compare of the baseline in the dataset. I wonder what is the reason? is there ablation study or any other results on this dataset?

---

> ### Author Response · Authors · 2018-11-16
> **Response**
>
> Thank you for the comments and we are glad that you found the paper interesting. We address the concerns you raised below.
>
> 1: While many works have been published with a similar theme (many of them we mention and cite in the paper), this specific configuration is novel, as well as the application to RL. Regarding the differences with MAC (which is indeed close to our own architecture) we replied this to reviewer 2 as well:
>
> In terms of differences, first we would say that the existence of the question as guidance to the controller in MAC is a fundamental difference mainly because it serves as a very strong top-down bias for the vision (read) module - it’s not just in the initialization of the reasoning, but throughout the whole process. In the RL case you could think of the reward as having a somewhat related role, but we feel this is a big difference. Second, our model is trained end-to-end so the vision can adapt and produce keys/values which are task relevant - MAC uses a pre-trained ImageNet module (though we don’t argue MAC can’t be learned end-to-end, it’s just that it wasn’t, and in the RL case, it’s not clear what pre-trained network would be useful). Lastly, MAC does not use a spatial basis of any sort, so it can’t reason about absolute positioning as well as limited relative positioning (trip wires, for example, wouldn’t be possible without this).
>
> 2: With NLP it is common to have the output of the attention model to be the same size as the input (because of the self-attention, all-to-all connectivity). This is not the case in this model since we have a severe spatial bottleneck from image size to single vector size. It’s not clear how to build a multilayer attention model like this, but this is certainly an interesting research direction.
>
> 3: We believe the issue here is not overfitting but rather optimization difficulties. It stems from running a high-capacity convnet over many time steps and propagating gradients. A new model trained with sequence length 8 achieves Top-1 70.1%, Top-5 88.6%. We have recently found a curriculum helps as well; a model that is first trained with sequence length 4 for 2e5 iterations and then further trained with sequence length 8 achieves Top-1 71.7%, Top-5 89.8% on sequence length 8.
>
> EDIT: We have a quick update regarding 3. A new model, also trained with a curriculum where we switch from sequence length 4 to 8 at 1e5 iterations, achieves Top-1 74.5%, Top-5 91.5%. This is now better than our sequence length 4 model and also our baseline.
>
> 4: We have found that our method provides a more focused indication of the salient areas in the image for the agent.  In section 4.1.6, we compare with a saliency mapping method that blurs specific regions of the input and measures the change of the policy or value function.  Compared to this, our method produces sharper attention maps and more naturally matches the structure of the image (the alternate analysis can only produce roughly circular regions of interest).
> We have investigated the applications of GradCAM to our baseline model on ATARI.  This produces interesting visualizations and is more adapted to the structure of the problem than the saliency visualization.  The biggest advantage our model has over GradCAM is it shows all the information that the agent is extracting from the image, not just what is relevant to produce a certain action.  GradCAM is a very powerful tool for showing why a certain action is being taken, but it does not show the full information the agent is extracting from the environment.  Especially when there is an LSTM above the convolutional stack, information may be extracted at the current timestep that won’t be used until several timesteps in the future, and so would not show up in a GradCAM visualization of the current frame.  One could imagine a modification of the GradCAM architecture which analyzes the gradient of the conv layer with respect to some future action, but this may quickly begin to overwhelm our ability to interpret the images.  Using our method, the agent cannot extract any information from the current frame that is not attended to, so we are sure we have a full picture of what is relevant for either the current action or any future action.
>
> 5: We didn’t focus too much on metrics for kinetics; we were more interested in understanding the attention maps qualitatively. The model has optimization issues on long time sequences, as seen in Imagenet sequence size 8.

---

> > ### Author Response · Authors · 2018-11-19
> > **new results for imagenet**
> >
> > EDIT: We have a quick update regarding 3. A new model, also trained with a curriculum where we switch from sequence length 4 to 8 at 1e5 iterations, achieves Top-1 74.5%, Top-5 91.5%. This is now better than our sequence length 4 model and also our baseline.

---

### Meta-Review · Area_Chair1 · 2018-12-13
**good analysis of the proposed method; lacks novelty**

**Confidence:** 4
**Recommendation:** Reject

**Metareview:**

1. Describe the strengths of the paper.  As pointed out by the reviewers and based on your expert opinion.

The paper
- tackles an interesting problem
- makes a concerted effort to provide qualititative results that give insight into the models behaviour.
- sufficiently cites related work.

2. Describe the weaknesses of the paper. As pointed out by the reviewers and based on your expert opinion. Be sure to indicate which weaknesses are seen as salient for the decision (i.e., potential critical flaws), as opposed to weaknesses that the authors can likely fix in a revision.

- The model architecture lacks novelty.
- There was also agreement that the contributions - (i) minor modifications of existing sequential attention-based models, and (ii) application to the RL domain - are minor.
- A lot of space in the paper (section 4.2) is devoted to exploring the use of this model for image classification and video action recognition. However the proposed model performed poorly compared to SOTA methods for this task and no motivation was given for why the proposed model would be useful for such tasks.

All three points impacted the final decision.

3. Discuss any major points of contention. As raised by the authors or reviewers in the discussion, and how these might have influenced the decision. If the authors provide a rebuttal to a potential reviewer concern, it’s a good idea to acknowledge this and note whether it influenced the final decision or not. This makes sure that author responses are addressed adequately.

There was high agreement between the reviewers on the main drawbacks of the paper, before and after the rebuttal.
The AC considered the rebuttals by the authors (in which they argued that there was sufficient contribution) but, in the end, agreed with the reviewers' assessments.

4. If consensus was reached, say so. Otherwise, explain what the source of reviewer disagreement was and why the decision on the paper aligns with one set of reviewers or another.

The reviewers reached a consensus that the paper should be rejected.